# Cancer symptom experience and help-seeking behaviour during the COVID-19 pandemic in the UK: a cross-sectional population survey

Harriet D Quinn-Scoggins,[1] Rebecca Cannings-John,[2] Yvonne Moriarty [ID],[2] Victoria Whitelock,[3] Katriina L Whitaker,[4] Detelina Grozeva,[2] Jacqueline Hughes,[2] Julia Townson,[2] Kirstie Osborne,[3] Mark Goddard,[2] Grace M McCutchan [ID],[5] Jo Waller [ID],[6] Michael Robling [ID],[2,7] Julie Hepburn,[8] Graham Moore,[7] Ardiana Gjini,[9,10] Kate Brain [ID] [1]

HDQ-S and RC-J are joint first authors.

For numbered affiliations see end of article.

**Correspondence to**
Dr Harriet D Quinn-Scoggins; Quinn-ScogginsHD@cardiff.ac.uk

## ABSTRACT

**Objectives** To understand self-reported potential cancer symptom help-seeking behaviours and attitudes during the first 6 months (March–August 2020) of the UK COVID-19 pandemic.

**Design** UK population-based survey conducted during August and September 2020. Correlates of help-seeking behaviour were modelled using logistic regression in participants reporting potential cancer symptoms during the previous 6 months. Qualitative telephone interviews with a purposeful subsample of participants, analysed thematically.

**Setting** Online UK wide survey.

**Participants** 7543 adults recruited via Cancer Research UK online panel provider (Dynata) and HealthWise Wales (a national register of 'research ready' participants) supplemented with social media (Facebook and Twitter) recruitment. 30 participants were also interviewed.

**Main outcome measures** Survey measures included experiences of 15 potential cancer symptoms, help-seeking behaviour, barriers and prompts to help-seeking.

**Results** Of 3025 (40.1%) participants who experienced a potential cancer symptom, 44.8% (1355/3025) had not contacted their general practitioner (GP). Odds of help-seeking were higher among participants with disability (adjusted OR (aOR)=1.38, 95% CI 1.11 to 1.71) and who experienced more symptoms (aOR=1.68, 95% CI 1.56 to 1.82), and lower among those who perceived COVID-19 as the cause of symptom(s) (aOR=0.36, 95% CI 0.25 to 0.52). Barriers included worries about wasting the doctor's time (1158/7543, 15.4%), putting strain on healthcare services (945, 12.6%) and not wanting to make a fuss (907, 12.0%). Interviewees reported reluctance to contact the GP due to concerns about COVID-19 and fear of attending hospitals, and described putting their health concerns on hold.

**Conclusions** Many people avoided healthcare services despite experiencing potential cancer symptoms during the COVID-19 pandemic. Alongside current help-seeking campaigns, well-timed and appropriate nationally coordinated campaigns should signal that services are open safely for those with unusual or persistent symptoms.

**Trial registration number** ISRCTN17782018.

## STRENGTHS AND LIMITATIONS OF THIS STUDY

⇒ To our knowledge, this is the first UK population survey of the impact of COVID-19 on help-seeking for potential cancer symptoms.
⇒ A large sample was recruited across two online surveys and data pooled where applicable, providing a larger dataset for analysis which was broadly representative of the UK population.
⇒ Data collection occurred between August and September 2020 and thus on the first lockdown period in the UK.
⇒ We assessed self-report of actual symptoms experienced during the first 6 months of the pandemic, reducing the known biases associated with retrospective recall of symptoms in patient samples or anticipated responses to hypothetical symptoms in community samples.
⇒ Survey data were supplemented with in-depth qualitative interviews, providing rich insight and context regarding symptom help-seeking behaviour during the pandemic.

## BACKGROUND

Cancer is the leading cause of mortality in the UK[1] and globally.[2] In countries with a 'gate-keeper' healthcare system such as the UK, most cancers are diagnosed symptomatically through primary care.[3] Diagnosing symptomatic cancer earlier can enable more timely treatment with better clinical outcomes across a range of cancers.[4 5] However, this route to cancer early diagnosis has been severely disrupted during the COVID-19 pandemic. Large reductions in demand for primary care services were noted[6] and estimates suggest that there were >380 000 fewer urgent suspected cancer referrals in the UK between March 2020 and March 2021, a reduction of approximately 13% compared with prepandemic

levels (CRUK Cancer Intelligence Team, Evidence of the impact of COVID-19 across the cancer pathway: Key stats, 2021). Modelling of cancer diagnostic delays in England estimate a substantial increase in the number of avoidable cancer deaths over the next 5 years due to the COVID-19 pandemic.[7] This has led to concerns that members of the public may not be coming forward to their general practitioner (GP) with potential cancer symptoms due to factors including fear of COVID-19 infection and concerns about placing additional burden on the National Health Service (NHS).[8]

During the first UK lockdown from March 2020, the UK government message to '*stay home, protect the NHS, save lives*' was intended to control the spread of COVID-19, but potentially sent a strong signal to the public that cancer can wait.[9] Consequently, the pandemic is likely to have affected key stages across the cancer diagnostic pathway[10] including the patient interval.[11] As set out in the Model of Pathways to Treatment,[12] the patient interval combines the time between a person noticing a bodily change or symptom to perceiving a reason to seek medical help (the appraisal interval), and the time between perceiving a reason to seek medical help to first contact with a medical professional (the help-seeking interval). In UK studies conducted before the COVID-19 pandemic, rates of self-reported symptom help-seeking in adults aged over 50 years ranged from 26.5% seeking help from their GP for at least one potential cancer symptom over a 1 month period,[13] to 60% over 12 months[13] and 67% over 3 months.[14] Adverse impact of the pandemic on people's willingness to seek help for potential cancer symptoms seems likely, especially for non-specific or respiratory symptoms that are similar to COVID-19 symptoms such as a persistent or changing cough, fatigue and breathlessness. Evidence from pre-COVID studies suggests that non-specific symptoms such as those previously mentioned may be overlooked or dismissed,[15] in part due to worry about wasting the doctor's time.[16] In adults with existing respiratory and cardiac comorbid conditions, potential cancer symptoms may be misattributed and not acted on.[17] Fear of COVID-19 infection may also deter attendance in healthcare settings, especially among high-risk and shielding groups.[18] Changes to healthcare service delivery during the pandemic, including remote GP consultations, may create additional barriers to accessing services.[19] We therefore anticipated that the UK adult population would be more reluctant to seek help for potential cancer symptoms than before the pandemic.

Evidence is needed regarding public perceptions of potential cancer symptoms and symptom help-seeking behaviour, and potential inequalities in help-seeking, to understand the factors driving reduced primary care service use in the UK during COVID-19. We conducted a large-scale population survey to examine self-reported symptom help-seeking attitudes and behaviour in a UK adult cohort during the pandemic. Selection of survey measures and framing of qualitative interview topics were guided by relevant health psychology theories including

the Model of Pathways to Treatment[11] and Common Sense Model of Self-Regulation.[20] In addition, we compared the overall proportion of participants seeking help during the first pandemic wave with UK prepandemic data reported in the Understanding Symptom Experiences Fully (USEFUL) study.[13]

## METHODS

### Study design

A prospective, mixed-methods observational cohort study in the UK population during the COVID-19 pandemic. The study protocol and analysis plans were preregistered on Open Science Framework.[21] Findings are reported in accordance with the Strengthening the Reporting of Observational Studies in Epidemiology guidelines for surveys and observational studies.[22 23]

### Survey participants and procedures

Two cross-sectional online surveys were conducted in parallel, the COVID-19 Health and Help-Seeking Behaviour Study (CABS) and the Cancer Research UK (CRUK) COVID-19 Cancer Awareness Measure (COVID-CAM). COVID-CAM was based on CRUK's Cancer Awareness Measure 2019.[24 25] Key measures were aligned where possible across the two surveys, and data pooled where appropriate. Eligible participants were aged 18 years or over (due to collecting additional survey data on attitudes and behaviours relating to cancer prevention and cervical screening), resident in the UK and able to speak English. Data were collected between 6 August and 18 September 2020, after the first UK lockdown which started on 23 March 2020. Study information was available online prior to participants providing electronic informed consent online.

Participants were recruited to the CABS survey via HealthWise Wales (HWW, a national register of 'research ready' participants)[26] and social media (Facebook and Twitter). Potentially under-represented groups including men, smokers, black, Asian and minority ethnic groups and people living in socioeconomically deprived areas were approached by HWW using personalised emails and Facebook-targeted advertising. Participants were recruited to the COVID-CAM survey via Dynata, an online panel provider (www.dynata.com). Quotas were placed on age, gender, social grade and UK region to recruit a nationally representative sample and sample size for ethnic minority groups was increased (relative to UK population statistics) to increase representation.

### Patient and public involvement

Patient and public involvement (PPI) was included at all stages from conceptualisation through to data interpretation. Working alongside CRUK's Cancer Insights Panel, the Wales Centre for Primary and Emergency Care Research PPI Group (Service Users for Primary and Emergency Care Research Group) and our study PPI co-applicant (JHep), all public-facing materials including study

information, consent procedures, survey and interview topic guides were reviewed and amended as appropriate. Our PPI co-applicant were also involved in results interpretation and how best to disseminate these to the wider population (video animation and infographic planned).

## Survey measures

Selection of measures was guided by clinical and academic expertise from the study management group, including our PPI groups and PPI co-applicant. New COVID-19-specific survey items were tested with PPI group members for acceptability prior to inclusion in the survey.

Measures were obtained from all participants across both CABS and COVID-CAM surveys unless otherwise stated. Data only collected in CABS are denoted by '~' and in COVID-CAM by '*'. Where relevant, a 6-month time frame was selected to include the beginning of the first UK lockdown on 23 March 2020. Two attention check questions were included in both surveys.[27]

### Demographic and health-related factors

Participants were asked in which region of the UK they lived, their date of birth ~/age*, gender, ethnicity, marital relationship, highest educational qualification and whether they considered themselves to have a disability. Experience of cancer was recorded by asking participants if they, anyone in their family or any of their friends had cancer. Smoking status was captured as never, former or current smoker.

### Symptom experience

Participants were asked if they had experienced any of the following 15 symptoms over the past 6 months: a persistent change in bowel habits, a persistent change in bladder habits, tiredness all the time, persistent unexplained pain, unexplained weight loss, a change in the appearance of a mole, an unexplained lump or swelling, unexplained bleeding, a persistent difficulty swallowing, a sore that does not heal, coughing up blood, shortness of breath, persistent hoarseness, a persistent cough, a change in an existing cough. The symptoms included were based on those in Connor et al[28] and included a range of non-specific, red flag and lung-specific symptoms. Response options were yes, no or prefer not to say.[15]

### Symptom help-seeking

For each symptom experienced, participants were asked "*How long after you first noticed the symptom did you contact the GP about it?*" Response options included: did not contact the GP; not contacted the GP yet but plan to; within 1 week of noticing the symptom; within 2 weeks of noticing the symptom; within 1 month of noticing the symptom; within 6 weeks of noticing the symptom; within 3 months of noticing the symptom; within 6 months of noticing the symptom; prefer not to say.[15] The method of categorising symptom help-seeking was based on the USEFUL study.[13] Outcomes were dichotomised as 'contacted GP in the last 6 months' versus 'no contact' for individual symptoms. For the composite outcome of GP contact across all symptoms, the outcome was 'contacted GP in the last 6 months for at least one symptom' versus 'no contact for any symptoms'.

### Perceived symptom cause

Participants who had experienced any of the eight following symptoms were asked what they thought caused the symptom using free text[15]: tired all the time, an unexplained lump or swelling, unexplained bleeding, coughing up blood, shortness of breath, persistent hoarseness, a persistent cough, a change in an existing cough. Free-text responses were independently coded by HDQ-S, GMMcC and YM into attribution categories[15]: cancer suspicion, COVID-19 (physical), COVID-19 (psychological), physical (non-cancer), psychological, external/normalising, do not know, exclude. Following independent coding of the first 20% of the data, Cohen's Kappa was used to assess the degree of inter-rater reliability per symptom. Inter-rater reliability was high for all symptoms (>0.80),[29] so no adjustments were made. For the purposes of the current study, symptom attribution categories were merged to create two variables: perceived cancer attribution (cancer/not cancer) and perceived COVID-19 attribution (COVID-19/not COVID-19).[15]

### Symptom recognition

For all 15 potential cancer symptoms, participants were asked "*Which of the following, if any, do you think could be warning signs or symptoms of cancer?*" Response options were: yes, I think this could be a sign of cancer; no, I don't think this could be a sign of cancer; do not know; unsure.[25] Items were summed to create a total symptom recognition score ranging from 0 to 15.

### Barriers and prompts to help-seeking

Participants were asked to select as many as applied from a list of 19 barriers experienced the last time they considered seeking medical help (online supplemental table S1).[25 28] Examples of barriers include "I found it embarrassing talking about my symptoms", "I worried about wasting the health professional's time" and "I had symptoms that might have been related to coronavirus". Response options included 'nothing put me off/delayed me in seeking medical attention' and 'prefer not to say'. Participants were asked to select as many as applied from a list of 20 prompts* that played a role in their decision to see or speak to a medical professional about their health (online supplemental table S2).[25] Examples of prompts include "I had a symptom that I thought might be a sign of cancer", "I had a symptom that was unusual for me" and "I could have a remote consultation (eg, by phone, email or video call)". Response options included 'other', 'I have never sought medical attention', "I don't remember" and 'prefer not to say'.

### Attitudes towards medical help-seeking during the pandemic

Participants were asked to rate their agreement with three items derived from a Cancer Research UK survey[30]: "I am confident that I would be safe from coronavirus

if I needed to attend an appointment at a hospital"; "I am confident that I would be safe from a coronavirus if I needed to attend an appointment at my GP surgery"; "I am worried about delays to cancer tests and investigations caused by coronavirus". A 4-point Likert scale (where 1=strongly agree and 4=strongly disagree) was used to assess agreement with each statement, with additional options of 'do not know' and 'prefer not to say'.

### Qualitative interviews

Survey participants who consented to interview were purposively sampled from the CABS study cohort according to age, gender and symptom experience. Consent for interview and audio-recording was reconfirmed verbally. A semi-structured topic guide (online supplemental material S3) was used to explore participants' views on attending primary and secondary healthcare in light of the COVID-19 pandemic, contextual influences on help-seeking and strategies to encourage future help-seeking. We aimed to recruit 30 participants in order to gain an in-depth understanding of views, while considering purposeful sampling to provide a range of participant demographic characteristics and symptom experiences. Interview participants were reimbursed with a £20 voucher. Transcribed anonymised data were thematically analysed.[31] Inductive data-driven codes and a priori deductive theory-driven codes were used to extrapolate themes. NVivo 12 (QSR international) was used as an aide to data organisation. Data were coded by HDQ-S, JHu and YM with 20% independently dual coded.

### Sample size

The study was powered to examine the correlates symptom help-seeking in those who experienced one or more potential cancer symptoms using a multivariable logistic regression model containing 15 candidate predictors.[32] For an outcome proportion of 0.20, the max ($R^2_{cs}$) value is 0.63. If we assume, conservatively, that the model will explain 15% of the variability, the anticipated $R^2_{cs}$ value is 0.15×0.63=0.095. This indicated that at least 1345 responders were required, corresponding to 269 events and an event per predictor parameter of 17.93. Inflating the sample size based on an estimated 20% symptom prevalence within a 3-month period,[33] the final sample size required for the primary survey analysis was 6725.

### Statistical analysis

Analyses were conducted using SPSS V.25.0 and Stata V.16.0. Data were weighted to match the UK population profile on age, gender, ethnicity and country (ie, devolved nation) using English 2011 Census and Office for National Statistics midyear estimates. Cases with missing data were excluded on a per-analysis basis. Descriptive analyses were used to identify sample characteristics, prevalence of potential cancer symptoms, help-seeking prompts and barriers (including a total barriers score ranging from 0 to 17) and, among those who had experienced potential cancer symptoms, symptom perceptions and help-seeking

behaviour. Sample characteristics and symptom prevalence are presented unweighted and weighted. Due to similar estimates, subsequent analyses are presented as unweighted.

### Correlates of symptom help-seeking behaviour

Descriptive summary statistics and logistic regression models were used to estimate the prevalence and odds respectively of GP help-seeking in those who had experienced at least one symptom (compared with not seeking help for any of their symptoms). The following key factors were examined: age group, gender, ethnicity, country, region, education, smoking status, marital relationship, disability, cancer status (self, family and friends), perceived symptom causes (cancer or COVID-19), barriers towards medical help-seeking, confidence in attending hospital and GP, delays in test results and cancer symptom recognition. We additionally fitted multivariable regression models to explore the independent contribution of potential factors by including all factors as independent variables to account for potential confounding of crude associations by other variables. The study was designed to fit descriptive models, capturing the association between dependent and independent variables, rather than for prediction or causality. Multicollinearity between factors was assessed using the variance inflation factor (VIF) (VIF >4 warrants further investigation). Data are reported as crude and adjusted ORs with 95% CIs.

## RESULTS

### Characteristics of participants

A total of 8167 participants responded to the survey in August and September, of whom 7543 (92.4%) were included (figure 1). Demographic characteristics of the pooled sample (n=7543) and by recruitment route are shown in table 1. Almost half the unweighted pooled sample was aged 55 years and over (n=3574, 47.4%) and female participants (3709, 49.2%). Most were of white ethnic background (6685, 88.6%) and living in England (4904, 65.0%). Over one-third had university-level education or higher (2892, 38.3%) and around two-thirds were married or cohabiting (4864, 64.5%). Current smokers and former smokers comprised 18.8% (1417) and 32.3% (2435) of the sample, respectively. Under a fifth (1284, 17.4%) reported having a disability and 8.7% (657) had experienced cancer themselves.

### Symptom prevalence

During the past 6 months, 40.1% (3025/7543) of survey participants had experienced at least one potential cancer symptom (table 2). Of these, a median of two symptoms per participant was reported (range 1–15 symptoms), while 31.8% (961/3025) experienced three or more symptoms. Nearly one-third of all participants had experienced at least one non-specific symptom (2284, 30.3%), almost a fifth reported at least one red flag symptom (1327, 17.6%), and at least one symptom possibly indicative

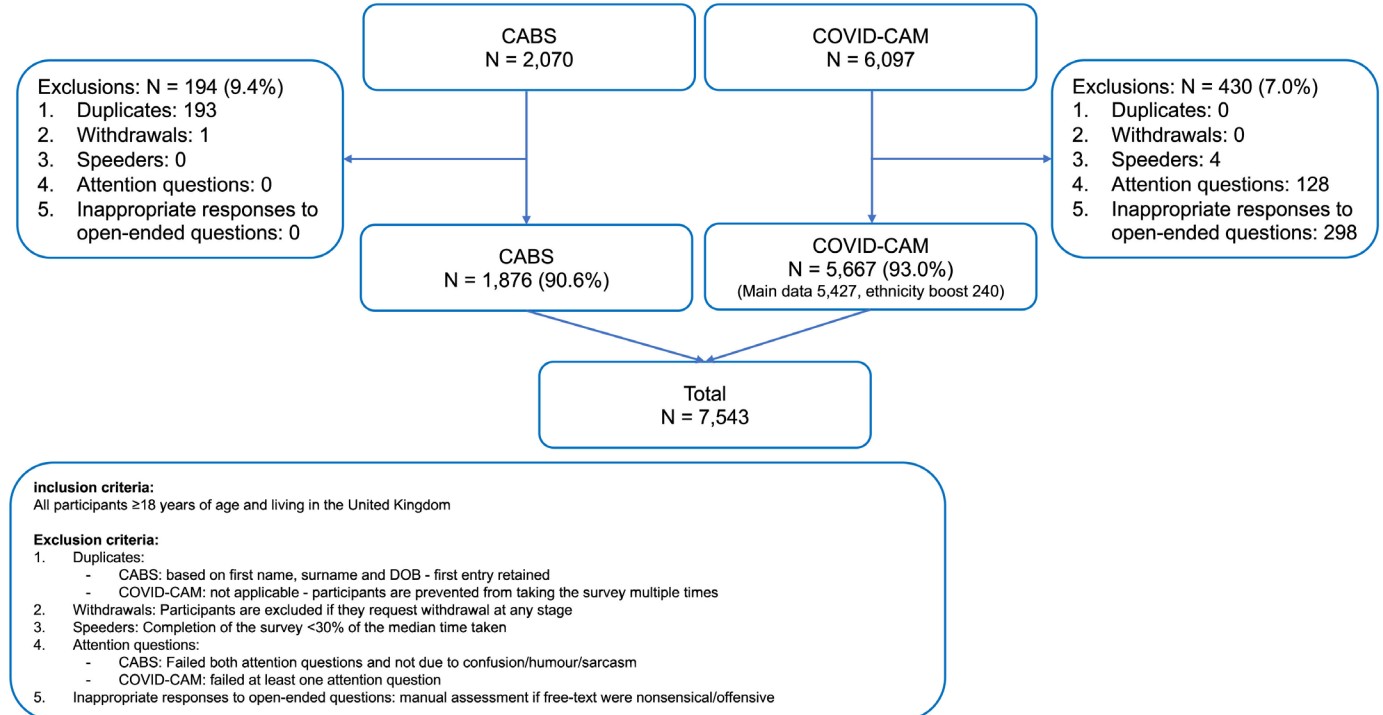

**Figure 1** Recruitment flow chart. CABS, COVID-19 Health and Help-Seeking Behaviour Study; COVID-CAM, Cancer Research UK's COVID-19 Cancer Awareness Measure; DOB, date of birth.

of lung cancer (1386, 18.4%). The prevalence of individual symptoms ranged from 21.3% (1603) ('tired all the time') to 1.5% (114) ('coughing up blood'). Among those reporting that they were 'tired all the time', had 'a persistent cough' or 'shortness of breath', around half said the symptom predated the pandemic (826/1603, 51.5%, 219/444, 49.3% and 510/1052, 48.5%, respectively) (online supplemental table S4).

## Symptom help-seeking

Among 3025 participants who experienced at least one potential cancer symptom, 44.8% (1355/3025) had not contacted the GP for any of their reported symptoms over a 6-month time frame, whereas 40.5% (3974/9810) had not contacted their GP over a 12-month time frame in the USEFUL study (table 2). A small proportion preferred not to say across all symptoms (1.1%). The proportion of participants not seeking help varied by symptom. A substantial proportion of participants had not sought help for red flag symptoms including coughing up blood (35/114, 30.7%), an unexplained lump or swelling (173/422, 41.0%) or a change in the appearance of a mole (229/391, 58.6%). Almost half of those who reported non-specific symptoms including 'a persistent change in bowel habits' (267/541, 49.4%) and 'a persistent change in bladder habits' (216/450, 48.0%) had not sought help from their GP, while a higher proportion (1031/1603, 64.3%) reporting being 'tired all the time' had not sought help. Around half of those experiencing lung-specific symptoms such as 'a persistent cough' (209/444, 47.1%) and 'shortness of breath' (538/1052, 51.1%) had not sought help. A further breakdown of help-seeking

according to recommended intervals is provided in online supplemental table S4.

As shown in table 2, the proportion of participants who had not contacted their GP over a 6-month time frame appeared to be higher than USEFUL study data for individual symptoms over a 12-month time frame including 'tired all the time' (64.3% (1031/1603) in the current study vs 57.8% (1778/3078) in the USEFUL study, 'unexplained weight loss' (51.9% (205/395) vs 44.6% (152/341)) and to a lesser extent 'shortness of breath' (51.1% (538/1052) vs 46.4% (1228/2647)). Proportions not seeking help for 'persistent change in bowel habits' (49.4% (267/541) vs 51.5% (682/1323)) and 'persistent cough' (47.1% (209/444) vs 49.7% (1088/2189)) appeared comparable. The proportion of participants who had not contacted their GP in the current study appeared to be lower than USEFUL study data for 'persistent difficulty swallowing' (40.9% (97/237) vs 63.0% (557/884)), 'persistent hoarseness' (47.5% (95/200) vs 71.3% (941/1319)), 'change in an existing cough' (42.9% (84/196) vs 51.3% (153/298)) and to a lesser extent 'coughing up blood' (30.7% (35/114) vs 34.1% (31/91)). It should be noted that relatively small numbers of participants in the current study reported experiencing the latter four symptoms.

## Correlates of symptom help-seeking

In unadjusted analyses, seeking help from the GP for at least one symptom was associated with former or current smoking, disability, experience of cancer (self), perceiving cancer as the cause of symptom(s) experienced and reporting a greater number of potential

**Table 1** Sample characteristics

| | Pooled sample n=7543 | Pooled sample weighted* n=7543 | CABS n=1876 | COVID-CAM n=5667 |
|---|---|---|---|---|
| **Age (years)** | | | | |
| 18–24 | 543 (7.2) | 665 (8.8) | 12 (0.6) | 531 (9.4) |
| 25–34 | 945 (12.5) | 1345 (17.8) | 53 (2.8) | 892 (15.7) |
| 35–44 | 1149 (15.2) | 1420 (18.8) | 132 (7.0) | 1017 (17.9) |
| 45–54 | 1221 (16.2) | 1420 (18.8) | 202 (10.8) | 1019 (18.0) |
| 55–64 | 1282 (17.0) | 1194 (15.8) | 417 (22.2) | 865 (15.3) |
| 65–74 | 1795 (23.8) | 816 (10.8) | 738 (39.3) | 1057 (18.7) |
| 75+ | 497 (6.6) | 590 (7.8) | 271 (14.4) | 226 (4.0) |
| Missing/Other/Prefer not to say | 111 (1.5) | 93 (1.2) | 51 (2.7) | 60 (1.1) |
| **Gender** | | | | |
| Male | 3807 (50.5) | 3681 (48.8) | 1044 (55.7) | 2763 (48.8) |
| Female | 3709 (49.2) | 3832 (50.8) | 827 (44.1) | 2882 (50.9) |
| Non-binary, transgender female or other | 27 (0.4) | 29 (0.4) | 5 (0.3) | 22 (0.4) |
| **Ethnicity** | | | | |
| White | 6685 (88.6) | 6948 (92.1) | 1821 (97.1) | 4864 (85.8) |
| Mixed/Multiple ethnic groups | 143 (1.9) | 153 (2.0) | 19 (1.0) | 124 (2.2) |
| Asian/Asian British | 458 (6.1) | 274 (3.6) | 15 (0.8) | 443 (7.8) |
| Black/African/Caribbean/Black British | 154 (2.0) | 135 (1.8) | 14 (0.7) | 150 (2.6) |
| Other ethnic group | 96 (1.3) | 26 (0.3) | | 86 (1.5) |
| Prefer not to say | 7 (0.1) | 8 (0.1) | 7 (0.4) | NA |
| **Country/Region** | | | | |
| England | 4904 (65.0) | 6311 (83.7) | 76 (4.1) | 4828 (85.2) |
| Wales | 2045 (27.1) | 376 (5.0) | 1797 (95.8) | 248 (4.4) |
| Scotland | 456 (6.0) | 601 (8.0) | 19 (1.0) | 453 (8.0) |
| Northern Ireland | 105 (1.4) | 225 (3.0) | | 105 (1.9) |
| England | | | | |
| North East England | 265 (3.5) | 376 (5.0) | | 265 (4.7) |
| North West England | 621 (8.2) | 826 (11.0) | | 618 (10.9) |
| Yorkshire and Humberside | 479 (6.4) | 526 (7.0) | | 476 (8.4) |
| East Midlands | 417 (5.5) | 601 (8.0) | | 415 (7.3) |
| East Anglia | 503 (6.7) | 676 (9.0) | | 500 (8.8) |
| West Midlands | 513 (6.8) | 676 (9.0) | | 508 (9.0) |
| South East England | 830 (11.0) | 1052 (13.9) | 24 (1.3) | 806 (14.2) |
| South West England | 473 (6.3) | 601 (8.0) | 9 (0.5) | 464 (8.2) |
| London | 803 (10.6) | 977 (12.9) | 27 (1.4) | 776 (13.7) |
| Prefer not to say | 33 (0.4) | 30 (0.4) | 0 (0.0) | 33 (0.6) |
| **Highest level of education** | | | | |
| Degree or higher degree | 2892 (38.3) | 2713 (36.0) | 897 (47.8) | 1995 (35.2) |
| A levels or further education | 2447 (32.4) | 2537 (33.7) | 542 (28.9) | 1905 (33.6) |
| O levels/GCSEs | 1565 (20.7) | 1694 (22.5) | 268 (14.3) | 1297 (22.9) |
| No formal qualifications | 412 (5.5) | 390 (5.2) | 105 (9.6) | 307 (5.4) |
| Still studying | 81 (1.1) | 87 (1.2) | 9 (0.5) | 72 (1.3) |
| Prefer not to say | 74 (1.0) | 65 (0.9) | 26 (1.4) | 48 (0.8) |
| Other | 72 (1.0) | 55 (0.7) | 29 (1.6) | 43 (0.8) |

**Table 1** Continued

| | Pooled sample n=7543 | Pooled sample weighted* n=7543 | CABS n=1876 | COVID-CAM n=5667 |
|---|---|---|---|---|
| Smoking status | | | | |
| Never smoked | 3586 (47.5) | 3601 (47.7) | 842 (45.9) | 2744 (48.4) |
| Former smoker | 2435 (32.3) | 2157 (28.6) | 839 (44.7) | 1596 (28.2) |
| Current smoker | 1417 (18.8) | 1706 (22.6) | 150 (8.0) | 1267 (22.4) |
| Other/Prefer not to say | 105 (1.4) | 79 (1.0) | 45 (2.3) | 60 (1.1) |
| Marital relationship | | | | |
| Not married or cohabiting | 2632 (34.9) | 2750 (36.5) | 561 (29.9) | 1978 (36.4) |
| Married or cohabiting | 4864 (64.5) | 4760 (63.1) | 1302 (69.4) | 3417 (63.0) |
| Prefer not to say | 47 (0.6) | 33 (0.4) | 13 (0.7) | 32 (0.6) |
| Disability | | | | |
| No | 6079 (82.6) | 6136 (83.4) | 1445 (78.7) | 4634 (83.8) |
| Yes | 1284 (17.4) | 1223 (16.6) | 390 (21.3) | 894 (16.2) |
| Experience of cancer | | | | |
| No | 1745 (23.1) | 2000 (26.5) | 157 (8.3) | 1558 (28.0) |
| Yes, other (family and friends)† | 5141 (68.2) | 5029 (66.7) | 1460 (77.8) | 3681 (65.0) |
| Yes, self | 657 (8.7) | 512 (6.8) | 259 (14.9) | 398 (7.0) |

Data are n (%) and unweighted unless otherwise stated.
*All data are weighted to match the UK adult population on age, gender, ethnicity and country.
†Participants stated that cancer was experienced in friends and family only and not in self.
CABS, COVID-19 Health and Help-Seeking Behaviour Study cohort recruited via HealthWise Wales and social media; COVID-CAM, Cancer Research UK's COVID-19 Cancer Awareness Measure sample recruited via Dynata, an online panel provider; NA, not available as an option.

cancer symptoms (table 3). Perceiving COVID-19 as the cause of symptom(s) was associated with lower odds of help-seeking. There were no other statistically significant unadjusted associations. After adjustment for other factors, disability, reporting more symptoms and not perceiving COVID-19 as the cause of symptom(s) experienced remained statistically significantly associated with higher odds of help-seeking.

### Help-seeking attitudes, barriers and prompts

Of the overall sample (n=7543), around two-thirds reported feeling safe from COVID-19 if they needed to attend an appointment at their GP practice (5142, 68.2%) or hospital (4613, 61.2%). Nearly three-quarters (5452, 72.3%) were worried about delays to cancer tests and investigations due to COVID-19.

The most frequently endorsed barriers to medical help-seeking in the overall sample were worry about wasting the healthcare professional's time (1158, 15.4%), worry about putting extra strain on the NHS (954, 12.6%), not wanting to be seen as someone who makes a fuss (907, 12.0%), difficulty getting an appointment with a particular healthcare professional (774, 10.3%) and worry about catching COVID-19 (721, 9.6%). Remote consulting was one of the least frequently endorsed barriers (361, 4.8%) (online supplemental table S1). A median of one barrier (25th–75th centile 1–2 barriers, range 0–14) was identified per participant.

For COVID-CAM survey participants (n=5667), the main prompts to speaking to a medical professional were having a symptom that was bothersome (1008, 17.8%), did not go away (957, 16.9%), was painful (811, 14.3%) and unusual (706, 12.5%) and having a feeling that something was not right (721, 12.7%) (online supplemental table S2).

### Qualitative results

Thirty participants were interviewed postsurvey completion (September–November 2020). Just over half were male (n=17), had received a higher education qualification or degree (n=19), lived in Wales (n=25) and were from a white ethnic background (n=23). The average age was 55 years (range 26–76 years). Exemplary quotes are provided in table 4. Codes and code definitions identified relating to the key themes presented on symptom experiences, fear of help-seeking and experiences of help-seeking are provided in online supplemental table S5.

### Symptom experiences

Many participants reported noticing a change to their health or well-being during the 6 months from the start of the first UK lockdown. This was commonly attributed to changes in existing health conditions such as asthma or diabetes or side effects of medication. This was more notable for non-specific symptoms such as tiredness all the time. As a result, participants delayed their help-seeking,

**Table 2** Participants experiencing potential cancer symptoms and associated symptom help-seeking

| Potential cancer symptom | Had symptom* n/7543 (%) | Had symptom— weighted† n/7543 (%) | Did not contact GP in the last 6 months‡ n/S (%) | Did not contact GP in the last 12 months— USEFUL study§ n (%) | Contacted GP in the last 6 months¶ n/S (%) | Contacted GP in the last 12 months—USEFUL study§ n (%) |
|---|---|---|---|---|---|---|
| Non-specific symptom | | | | | | |
| A persistent change in bowel habits | 541 (7.2) | 525 (7.0) | 267 (49.4) | 682/1323 (51.5) | 254 (47.0) | 641/1,323 (48.5) |
| A persistent change in bladder habits | 450 (6.0) | 414 (5.5) | 216 (48.0) | – | 227 (50.4) | – |
| Tired all the time | 1603 (21.3) | 1614 (21.4) | 1031 (64.3) | 1778/3078 (57.8) | 540 (33.7) | 1300/3078 (42.2) |
| Persistent unexplained pain | 662 (8.8) | 646 (8.6) | 286 (43.2) | – | 361 (54.5) | – |
| Non-specific/red flag symptom | | | | | | |
| Unexplained weight loss | 395 (5.2) | 433 (5.7) | 205 (51.9) | 152/341 (44.6) | 179 (45.3) | 189/341 (55.4) |
| Red flag symptom | | | | | | |
| A change in the appearance of a mole | 391 (5.2) | 402 (5.3) | 229 (58.6) | – | 157 (40.2) | – |
| An unexplained lump or swelling | 422 (5.6) | 418 (5.5) | 173 (41.0) | – | 239 (56.6) | – |
| Unexplained bleeding | 267 (3.5) | 291 (3.9) | 115 (43.1) | – | 143 (53.6) | – |
| A persistent difficulty swallowing | 237 (3.1) | 248 (3.3) | 97 (40.9) | 557/884 (63.0) | 128 (54.0) | 327/884 (37.0) |
| A sore that does not heal | 291 (3.9) | 297 (3.9) | 146 (50.2) | – | 128 (44.0) | – |
| Red flag/Lung-specific symptom | | | | | | |
| Coughing up blood | 114 (1.5) | 127 (1.7) | 35 (30.7) | 31/91 (34.1) | 67 (58.8) | 60/91 (65.9) |
| Lung-specific symptom | | | | | | |
| Shortness of breath | 1052 (13.9) | 966 (12.8) | 538 (51.1) | 1228/2647 (46.4) | 484 (46.0) | 1419/2647 (53.6) |
| Persistent hoarseness | 200 (2.7) | 206 (2.7) | 95 (47.5) | 941/1319 (71.3) | 96 (48.0) | 378/1319 (28.7) |
| A persistent cough | 444 (5.9) | 401 (5.3) | 209 (47.1) | 1088/2189 (49.7) | 230 (51.8) | 1101/2189 (50.3) |
| A change in an existing cough | 196 (2.6) | 219 (2.9) | 84 (42.9) | 153/298 (51.3) | 100 (51.0) | 145/298 (48.7) |
| *All potential cancer symptoms* | *3025 (40.1)**\** | *2909 (38.6)* | *1355/3025 (44.8)††* | *3974/9810 (40.5)* | *1636/3025 (54.1)‡‡* | *5836/9810 (59.5)* |
| *Non-specific symptom* | *2284 (30.3)**\** | *2261 (30.0)* | | | | |
| *Red flag symptom* | *1327 (17.6)**\** | *1310 (17.4)* | | | | |
| *Lung-specific symptom* | *1386 (18.4)**\** | *1289 (17.1)* | | | | |

Data are n (%) and unweighted unless otherwise stated; N=number, N/S=number of respondents representing each symptom help-seeking behaviour/number of respondents who had this symptom.
*Denominator includes those who did not have a symptom and those who preferred not to say (around 1% of the sample).
†All data are weighted to match the UK adult population on age, gender, ethnicity and country.
‡Includes participants who had not contacted the GP yet, but planned to. 'Did not contact GP' and 'contacted GP' columns are mutually exclusive. Denominator includes participants who preferred not to say.
§Comparator data for adults aged >50 years who did and did not contact the GP in the last 12 months.[13]
¶A further breakdown of help-seeking intervals is in online supplemental table S4.
**At least one potential cancer symptom reported.
††Did not contact the GP for symptoms reported in the last 6 months. 'Did not contact GP' and 'contacted GP' columns are mutually exclusive. Denominator also includes 34 (1.1%) who preferred not to say across all their symptoms.
‡‡Contacted the GP for at least one symptom in the last 6 months.
GP, general practitioner.

or did not seek help at all, to avoid bothering the doctor when they assumed that they already knew the cause. Even when participants reported red flag symptoms, there was discussion of delaying due to concerns about the NHS being overstretched. Several participants described accessing other services as a way of easing

**Table 3** Unadjusted and adjusted logistic regression models for self-reported symptom help-seeking in participants who experienced at least one potential cancer symptom (n=3025*)

| | Did not contact GP† n=1355 | Contacted GP† n=1636 | Crude OR (95% CI) | Adjusted OR (95% CI) n=2281 |
|---|---|---|---|---|
| Age (years) n=2942 | | | | |
| 18–24 | 127 (46.4) | 147 (53.6) | ref (1.0) | ref (1.0) |
| 25–34 | 199 (49.7) | 201 (50.3) | 0.87 (0.64 to 1.19) | 0.80 (0.52 to 1.24) |
| 35–44 | 196 (45.7) | 233 (54.3) | 1.03 (0.76 to 1.39) | 1.12 (0.73 to 1.71) |
| 45–54 | 211 (47.1) | 237 (52.9) | 0.97 (0.72 to 1.31) | 1.11 (0.72 to 1.70) |
| 55–64 | 220 (45.2) | 267 (54.8) | 1.05 (0.78 to 1.41) | 1.08 (0.70 to 1.67) |
| 65–74 | 284 (41.8) | 396 (58.2) | 1.20 (0.91 to 1.60) | 1.29 (0.83 to 2.00) |
| 75+ | 97 (43.3) | 127 (56.7) | 1.13 (0.79 to 1.61) | 1.20 (0.72 to 2.00) |
| P value | | | 0.261 | 0.321 |
| Gender n=2978 | | | | |
| Male | 625 (43.7) | 804 (56.3) | ref (1.0) | ref (1.0) |
| Female | 727 (46.9) | 822 (53.1) | 0.88 (0.76 to 1.02) | 0.99 (0.82 to 1.21) |
| P value | | | 0.080 | 0.951 |
| Ethnicity n=2988 | | | | |
| White | 1193 (45.0) | 1457 (55.0) | ref (1.0) | ref (1.0) |
| Ethnic minorities‡ | 160 (47.3) | 178 (52.7) | 0.91 (0.73 to 1.14) | 0.85 (0.61 to 1.18) |
| P value | | | 0.420 | 0.328 |
| Country n=2971 | | | | |
| England | 854 (47.2) | 955 (52.8) | ref (1.0) | ref (1.0) |
| Wales | 405 (42.5) | 549 (57.5) | 1.21 (1.03 to 1.42) | 1.23 (0.98 to 1.54) |
| Scotland | 72 (43.1) | 95 (56.9) | 1.18 (0.86 to 1.62) | 1.35 (0.90 to 2.02) |
| Northern Ireland | 15 (36.6) | 26 (63.4) | 1.55 (0.82 to 2.95) | 1.79 (0.62 to 5.20) |
| P value | | | 0.062 | 0.140 |
| Country/Region§ n=2971 | | | | |
| Wales | 405 (42.5) | 549 (57.5) | ref (1.0) | |
| Scotland | 72 (43.1) | 95 (56.9) | 0.97 (0.70 to 1.36) | |
| Northern Ireland | 15 (36.6) | 26 (63.4) | 1.28 (0.67 to 2.55) | |
| England | | | | |
| North East England | 59 (53.6) | 51 (46.4) | 0.64 (0.43 to 0.95) | |
| North West England | 109 (45.2) | 132 (54.8) | 0.89 (0.67 to 1.19) | |
| Yorkshire and Humberside | 85 (47.0) | 96 (53.0) | 0.83 (0.61 to 1.15) | |
| East Midlands | 72 (50.3) | 71 (49.7) | 0.73 (0.51 to 1.03) | |
| South East England | 130 (45.8) | 154 (54.2) | 0.87 (0.67 to 1.14) | |
| East Anglia | 72 (43.1) | 95 (56.9) | 0.97 (0.70 to 1.36) | |
| South West England | 84 (48.0) | 91 (52.0) | 0.80 (0.58 to 1.10) | |
| West Midlands | 96 (46.6) | 110 (53.4) | 0.85 (0.62 to 1.14) | |
| London | 147 (48.7) | 155 (51.3) | 0.78 (0.60 to 1.01) | |
| P value | | | 0.379 | |
| Highest level of education n=2934 | | | | |
| Degree or higher degree | 514 (47.2) | 574 (52.8) | ref (1.0) | ref (1.0) |

Continued

**Table 3** Continued

| | Did not contact GP† n=1355 | Contacted GP† n=1636 | Crude OR (95% CI) | Adjusted OR (95% CI) n=2281 |
|---|---|---|---|---|
| A levels or further education | 460 (46.2) | 536 (53.8) | 1.04 (0.88 to 1.24) | 0.91 (0.73 to 1.14) |
| O levels/GCSEs | 265 (42.3) | 362 (57.7) | 1.22 (1.00 to 1.49) | 1.07 (0.83 to 1.39) |
| Still studying | 16 (36.4) | 28 (63.6) | 1.57 (0.84 to 2.93) | 1.21 (0.51 to 2.89) |
| No formal qualifications | 73 (40.8) | 106 (59.2) | 1.30 (0.94 to 1.79) | 0.77 (0.51 to 1.16) |
| P value | | | 0.127 | 0.494 |
| Smoking status n=2948 | | | | |
| Never smoked | 595 (50.6) | 580 (49.4) | ref (1.0) | ref (1.0) |
| Former smoker | 436 (41.2) | 623 (58.8) | 1.47 (1.24 to 1.73) | 1.16 (0.94 to 1.44) |
| Current smoker | 302 (42.3) | 412 (57.7) | 1.40 (1.16 to 1.69) | 1.03 (0.80 to 1.32) |
| P value | | | <0.001 | 0.358 |
| Marital relationship n=2976 | | | | |
| Not married or cohabiting | 516 (46.4) | 597 (53.6) | ref (1.0) | ref (1.0) |
| Married or cohabiting | 831 (44.6) | 1032 (55.4) | 1.07 (0.92 to 1.25) | 0.95 (0.78 to 1.16) |
| P value | | | 0.352 | 0.647 |
| Disability n=2900 | | | | |
| No | 1042 (50.7) | 1014 (49.3) | ref (1.0) | ref (1.0) |
| Yes | 281 (33.3) | 563 (66.7) | 2.06 (1.74 to 2.43) | 1.38 (1.11 to 1.71) |
| P value | | | <0.001 | 0.003 |
| Experience of cancer n=2991 | | | | |
| No | 270 (50.3) | 267 (49.7) | ref (1.0) | ref (1.0) |
| Yes, other (family and friends)¶ | 974 (45.8) | 1154 (54.2) | 1.20 (0.99 to 1.45) | 1.09 (0.84 to 1.43) |
| Yes, self | 111 (34.0) | 215 (66.0) | 1.96 (1.47 to 2.60) | 1.12 (0.76 to 1.66) |
| P value | | | <0.001 | 0.783 |
| Symptom attributed to cancer¶ n=2990 | | | | |
| Not cancer | 1342 (45.6) | 1601 (54.4) | ref (1.0) | ref (1.0) |
| Cancer | 12 (25.5) | 35 (74.5) | 2.44 (1.26 to 4.73) | 1.30 (0.56 to 3.04) |
| P value | | | 0.008 | 0.547 |
| Symptom attributed to COVID¶ n=2990 | | | | |
| Not COVID | 1214 (44.0) | 1547 (56.0) | ref (1.0) | ref (1.0) |
| COVID | 140 (61.1) | 89 (38.9) | 0.50 (0.38 to 0.66) | 0.36 (0.25 to 0.52) |
| P value | | | <0.001 | <0.001 |
| Number of barriers to help-seeking (0–17) n=2991 Median (25th–75th centiles) | 1 (1 to 3) | 1 (1 to 3) | 1.04 (1.00 to 1.08) | 0.97 (0.92 to 1.03) |
| P value | | | 0.057 | 0.315 |
| Confident that I would be safe from coronavirus if I needed to attend an appointment at a hospital n=2645 | | | | |
| Strongly agree | 251 (44.7) | 311 (55.3) | ref (1.0) | ref (1.0) |
| Somewhat agree | 518 (44.3) | 650 (55.7) | 1.01 (0.83 to 1.24) | 0.86 (0.64 to 1.15) |

Continued

**Table 3** Continued

| | Did not contact GP† n=1355 | Contacted GP† n=1636 | Crude OR (95% CI) | Adjusted OR (95% CI) n=2281 |
|---|---|---|---|---|
| Somewhat disagree | 268 (44.0) | 341 (56.0) | 1.03 (0.82 to 1.29) | 0.74 (0.51 to 1.06) |
| Strongly disagree | 149 (48.7) | 157 (51.3) | 0.85 (0.64 to 1.12) | 0.58 (0.36 to 0.94) |
| P value | | | 0.547 | 0.150 |
| Confident that I would be safe from coronavirus if I needed to attend an appointment at my GP surgery n=2692 | | | | |
| Strongly agree | 337 (47.3) | 375 (52.7) | ref (1.0) | ref (1.0) |
| Somewhat agree | 545 (44.1) | 690 (55.9) | 1.14 (0.95 to 1.37) | 1.21 (0.92 to 1.58) |
| Somewhat disagree | 217 (41.1) | 311 (58.9) | 1.29 (1.03 to 1.62) | 1.47 (1.02 to 2.12) |
| Strongly disagree | 102 (47.0) | 115 (53.0) | 1.01 (0.75 to 1.37) | 0.93 (0.55 to 1.56) |
| P value | | | 0.146 | 0.082 |
| Worried about delays to cancer tests and investigations caused by COVID-19 n=2720 | | | | |
| Strongly agree | 479 (43.8) | 614 (56.2) | ref (1.0) | ref (1.0) |
| Somewhat agree | 534 (46.2) | 621 (53.8) | 0.91 (0.77 to 1.07) | 1.03 (0.83 to 1.25) |
| Somewhat disagree | 126 (41.0) | 181 (59.0) | 1.12 (0.87 to 1.45) | 1.18 (0.86 to 1.62) |
| Strongly disagree | 77 (46.7) | 88 (53.3) | 0.89 (0.64 to 1.24) | 0.97 (0.65 to 1.45) |
| P value | | | 0.340 | 0.762 |
| Cancer symptom recognition score (score 0–15) n=2991 Median (25th–75th centiles) | 11 (8 to 14) | 11 (8 to 14) | 1.00 (0.99 to 1.02) | 1.01 (0.99 to 1.04) |
| P value | | | 0.789 | 0.263 |
| Number of symptoms (maximum 15) n=2991 Median (25th–75th centiles) | 1 (1 to 2) | 2 (1 to 4) | 1.62 (1.53 to 1.72) | 1.68 (1.56 to 1.82) |
| P value | | | <0.001 | <0.001 |

Data are n (%) and unweighted unless otherwise stated.

An OR >1 indicates increased odds of help-seeking.

*n=34 participants indicated that they prefer not to say across all symptoms and were excluded from the analysis.

†Did not contact the GP for symptoms reported in the last 6 months/contacted the GP for at least one symptom in the last 6 months.

‡Ethnicity groups combined for analysis due to small numbers: 'mixed/multiple ethnic groups', 'Asian/Asian British', 'black/African/Caribbean/black British', 'other ethnic group', 'prefer not to say'.

§Not included in multivariable model due to collinearity with country.

¶Participants stated that cancer was experienced in friends and family only and not in self.

GP, general practitioner; ref, reference.

pressures on their GP practice, for example, by phoning 111 or contacting their pharmacist. When making decisions about help-seeking, participants weighed the risks of their clinical need against the risks of catching or exposing others to COVID-19 and burdening the NHS. Some participants conveyed the sentiment that the least they could do to help was to stay away from the NHS.

### Fear of help-seeking

All participants expressed fear or nervousness about presenting to primary or secondary care. For some, levels of fear were very high. This was commonly associated with 'the unknown' and potentially encountering other members of the public who may not adhere to social distancing guidance. These acted as barriers to timely medical help-seeking. Changes to GP practice procedures invoked worry and hesitancy due to not knowing or understanding the new measures. Examples included the use of new online and telephone triage systems and one-way systems in medical buildings. Participants understood the need for these adaptations, although felt that more support could be provided on how to navigate these changes. Participants expressed particular concern for patients with low digital literacy and those with English as a second language or additional mobility needs.

**Table 4**  Exemplary participant quotes by major theme for symptom experiences, fear of help-seeking and experiences of help-seeking

| Major theme | Exemplary participant quotes (participant ID, gender, age (years), nation of residency) (quotes provided in intelligent verbatim) |
|---|---|
| Symptom experiences | "P: No, apart from the return of the backache … but I think I know why that is, so I haven't done anything about it. Because I know what's going to help it, so as soon as I can go back to the gym, or decide to go back to the gym and start those classes, it will be fine". (64021806, female, 64, Wales) |
|  | "P: I noticed I was getting increasingly tired… I had a couple of other symptoms as well, which made me think my Levothyroxine dose was now insufficient". (63984720, male, 62, Wales) |
|  | "I: Okay and has the pandemic affected or changed how you think about doctors' visits and appointments at all?<br>P: I would certainly said I've been more reluctant, I would have stayed away and just dealt with it, rather than perhaps going to see a doctor at an early stage". (64948240, female, 46, Wales) |
|  | "P:… over the weekend I had a, second time in my life, a bad migraine, and thankfully I'm feeling better but I had thought to myself at what point am I going to go to the GP about not feeling better. And will I… you know am I less likely to go because they're under strain? And I probably am a bit less likely to go, delay it a little bit longer". (64078317, female, 46, England) |
|  | "P: … it's certainly changed my mind because like I say I'm of the mindset that says if it's not sort of life threatening critical then, you know, it can wait. So yes, you know I had a certainly different mentality and part of that I think is because of the strain that was put on the health service and all those within it initially that you perhaps didn't want to disturb them". (65205685, female, 63, Wales) |
| Fear of help-seeking | "P: … I haven't been there, the last time I went there, I think it was in the January when I had my annual COPD and CHD review… So, I hadn't been there since, and then I was reading all these horror stories, you know, the stuff we were seeing on the telly. You know the people were going into places, and they didn't even know they had the virus, they wasn't showing symptoms… And passing it on and I was thinking, this could happen to me in the doctor's surgery, but when I actually went to the surgery the whole layout had changed, it had all new furniture put in there, so it could be wiped down". (65205685, female, 63, Wales) |
|  | "P: Well if you're asking about hospital, I was supposed to go to hospital in lockdown see, but the thing is, I was too frightened because of Covid, I thought I'm not going to hospital. And I needed stitches in my knee, because I fell and I landed on both knees in the living room, I fell over the mat. I sliced my knee open, and I needed stitches bad, but I didn't go. My husband used butterfly stitches and done it that way. But I wouldn't go because of Covid see, because I was too frightened, because I didn't want to get Covid". (64018114, female, 44, Wales) |
|  | "P: … I mean my view to hospitals, prior to being in one myself, was that, you know there were people dying all over the place in every ward, every corridor with coronavirus. So yes, I would have been, as I say, certainly very cautious to have, to have wanted to put myself in that situation…. you know I was so impressed with how the hospital were operating when I was in there and, as I say if I'd had vision or understood what it was looking like, how it was working I probably wouldn't have had any concerns at all. I think the hospitals were the safest, safest place to be, is my view after the event, seeing how fantastically well the staff were, you know at following procedure etc… So yes if, you know, if you get that message across that, that a hospital, as I say, is probably the safest place than bloody Tesco's or the local pub or whatever. You know, you're very safe there". (65205685, female, 63, Wales) |
| Experiences of help-seeking | "P: … the surgery did a triage thing, the doctor called me and asked me to go and see them and that worked okay, you know, under the restrictions of the local GP, surgery, you know… They have, they've got, quite stringent processes… Yeah, I was content there, no serious misgivings, you accept their protocols and the new way of doing things and that was fine actually, no problem". (64026131, male, 62, Wales) |
|  | "P: Like I said that assumption a lot of people make as well… They assume that because you're okay, you're seeing them in real life, you're okay talking to them over the video, like I said I, I really don't feel comfortable using those video things. I can't sort of speak normally over them. I feel very disconnected from the person I just, I find it really hard to do". (64027453, male, 38, Wales) |
|  | "P: It has changed the whole system, you can't just make an appointment to go and see somebody, you have to go online, type in briefly what your problem is and then decide whether they call you back or whether they tell you what to do or whether they say I think we should meet face to face. Usually a telephone conversation first and then decide okay perhaps you'd better come down and see me. Which I did once… I think the system works very well actually.<br>I: Do you, so how does it compare then before the pandemic? Could you just make an appointment in those?<br>P: You could but it was always sort of three or four weeks ahead… With the new system, you seem to get some response within the next twenty-four hours which is a big improvement". (63986310, male, 76, Wales) |

I, interviewer; P, participant.

Fear of attending secondary care was acute for many. Some participants reported being too scared to attend secondary care appointments, treatments or procedures. They made this decision knowing that it could be detrimental to their health and well-being. However, those who did attend face-to-face in primary and/or secondary care described feeling 'safe' and 'secure' when attending. Participants expressed surprise that attending was at odds with their expectations of what it was going to be like. Participants described viewing 'scaremongering' media reports of hospitals being overrun with COVID-19 cases exacerbating their fears. Several participants were saddened that they had been manipulated by the media into feeling scared and avoiding healthcare, with consequences for their health.

## Experiences of help-seeking

When participants had contacted their GP, overall they were pleased with the quality of care received and the use of remote consultations. Some were hesitant about disclosing details of their health and medical history before a decision was made about whether they could speak to or see a doctor, feeling that this impacted on their privacy. The use of telephone consultations was praised by most who had received them. Many of these participants reported that it was easier and faster to get a GP appointment than before the pandemic, and that they would like to keep the change to remote consulting on the understanding that face-to-face appointments would be available based on clinical need.

## DISCUSSION

We conducted the first population study of cancer symptom experience and help-seeking behaviour during the COVID-19 pandemic in the UK. Among adults surveyed who experienced one or more potential cancer symptom during the first 6 months of the pandemic, nearly half had not sought help for any symptoms from the GP during this time, even for red flag symptoms. Reporting a disability and experiencing more symptoms were associated with higher odds of symptom help-seeking, whereas attributing a symptom(s) to COVID-19 was associated with lower odds. Qualitative data revealed reluctance to contact primary care services due to concerns about catching or transmitting COVID-19 and overburdening the NHS. Interviewees described delaying medical help-seeking due to fears that were driven by and exacerbated by media reports of COVID-19 in hospitals.

The prevalence of symptoms experienced over the 6-month period in the current study was in line with previous studies.[13 34] Symptom help-seeking behaviour during the first 6 months of the pandemic appeared to be lower than help-seeking reported in the USEFUL study over a 12-month time frame, overall and for individual symptoms such as persistent tiredness and unexplained weight loss, although direct comparison was restricted by methodological differences such as variation in symptom reporting time frames. Similarly to previous research, key help-seeking barriers in the current study included worry about wasting healthcare professionals' time, overstretching limited healthcare resources and accessing healthcare services (personal communication).[28 35] In a Spanish population sample, Petrova et al[36] also reported barriers to anticipated symptom help-seeking during the COVID-19 pandemic including worry about wasting the doctor's time and worry about what the doctor might find. International prepandemic research on barriers to help-seeking has found that UK adults are more likely to report worry about 'bothering the doctor' compared with those in other high-income countries.[16] Participants in our study described putting their health concerns on hold or self-managing conditions and concerns to avoid burdening

the NHS, suggesting a compounding of the 'British stiff upper lip' phenomenon observed in prepandemic research.[16] Novel COVID-specific barriers and attitudes reflecting concerns about COVID-19 infection in healthcare settings and delayed cancer testing were prevalent in both the current survey and qualitative interviews, but they did not contribute significantly to modelling help-seeking behaviour. Difficulty with remote healthcare consulting was not frequently endorsed; indeed, qualitative findings suggested that when participants had contacted their GP or visited hospital, they reported positive experiences that contrasted with their expectations. Retaining remote consultations alongside face-to-face consultations in future routine healthcare services was favoured.

The correlates of help-seeking behaviour in this study in part reinforce what has been observed in previous studies. The influence of disability and reporting more symptoms on help-seeking behaviour aligns with previous studies including Hannaford et al[13] in which people who were unable to work due to illness or disability were more likely to act on their symptoms. Mechanisms which serve to both increase and decrease timely presentation for symptoms have been previously identified and may vary by nature of comorbidity.[17 37] This relationship was observed in our qualitative interviews whereby participants who experienced a new or changing symptom attributed such changes to pre-existing conditions or medications, although results show this did not deter help-seeking in statistical analyses. In contrast, attributing symptoms to COVID-19 was associated with not contacting the GP and may have been influenced by government messaging to stay at home if experiencing any COVID-like symptoms. The decision not to act on symptoms experienced during the first UK pandemic wave may have been motivated by a desire to protect others in the community from COVID-19 infection, and to prevent healthcare services from being overwhelmed. A qualitative study of GPs' perceptions of changes in symptom help-seeking behaviour described patients as more vigilant about their health but also more reluctant to seek help as a result of the pandemic.[38] Our finding that current and former smokers were more likely to seek help was similar to findings reported in USEFUL study by Hannaford et al.[13] Although the association did not remain after adjustment in the present study, the consistency of this emerging finding with Hannaford et al warrants investigation in future research. It is possible, for example, that people who currently smoke or have previously smoked perceive an elevated personal risk status which may prompt symptom presentation. The total number of help-seeking barriers endorsed was not associated with help-seeking behaviour, and more fine-grained analysis of differentiated emotional, practical or service-related barriers is needed.

A key strength of our study was the focus on actual symptoms experienced during the last 6 months. This reduced the known biases associated with retrospective

recall of actual symptoms in patient samples or anticipated responses to hypothetical symptoms in community samples. Pooling data across two surveys provided a large sample that was broadly representative of the British population. However, we acknowledge that willingness and ability to complete an online survey was a prerequisite of study participation that may limit the generalisability of findings. Further methodological limitations are recognised, including the likelihood of reduced sample variation. Despite good representation of ethnic minority groups and people with lower education due to targeted recruitment, we did not observe differences in help-seeking previously identified among these groups.[25 30] This may reflect reduced statistical power to detect such effects because we restricted the analysis to actual symptom-helping among those who had experienced at least one potential cancer symptom. Further research is warranted to examine patterns of help-seeking for individual symptoms or subsets of symptoms (eg, respiratory) and receptiveness to remote GP consulting among participants with varying degrees of digital literacy and health motivation. We acknowledge the constraints on our ability to compare rates of symptom help-seeking during the pandemic with those reported prepandemic, due to methodological differences including the longer symptom reporting time frame (12 months) and older age inclusion criteria (>50 years) in the USEFUL study comparator. However, our qualitative findings indicate that people were not coming forward to their GP with symptoms during the first 6 months of the pandemic. The statistical modelling also showed that attributing symptoms to COVID-19 was associated with lower odds of help-seeking. This pattern may have contributed to the decline in GP referrals for suspected cancer that was observed during 2020.[6]

Evidence from this study highlights the need for continued investment in evidence-led, nationally funded and coordinated cancer awareness campaigns to legitimise seeking help for unusual or persistent symptoms. Clear, consistent information from a trusted source should encourage confidence in contacting the GP promptly, explain the changes to GP practice procedures and what to expect and alleviate worries about health service capacity and infection control in hospital settings. Credible patient stories with an emphasis on positive outcomes could be important in counteracting possible hyperbolic COVID-19 news reporting and to appropriately recontextualise accounts and support engagement with hospital outpatient appointments, treatments or investigations. Campaigns and other supporting activity could increase uptake and access to remote consulting as it becomes embedded in primary and secondary cancer care.[39] Evaluation of campaign activity and other interventions is essential to ensure that they reach diverse audiences and do not exacerbate inequalities. As the COVID-19 pandemic continues, research must continue to monitor the influences on help-seeking for potential cancer symptoms.

**Author affiliations**
[1]PRIME Centre Wales, Division of Population Medicine, School of Medicine, Cardiff University, Cardiff, UK
[2]Centre for Trials Research, Cardiff University, Cardiff, UK
[3]Cancer Intelligence, Cancer Research UK, London, UK
[4]School of Health Sciences, University of Surrey, Guildford, UK
[5]Division of Population Medicine, School of Medicine, Cardiff University, Cardiff, UK
[6]School of Cancer and Pharmaceutical Sciences, King's College London, London, UK
[7]DECIPHer (Centre for Development, Evaluation, Complexity and Implementation in Public Health Improvement), School of Social Sciences, Cardiff University, Cardiff, UK
[8]Public Involvement Community, Health and Care Research Wales Support Centre, Cardiff, UK
[9]Public Health Wales, Cardiff, UK
[10]Cardiff University, Cardiff, UK

**Acknowledgements** We are grateful to Cancer Research UK's Cancer Insights Patient Panel, Clinical Advisory Panel and GP Panel and PRIME Centre Wales SUPER Group for their helpful feedback. Cancer Research UK staff members and other researchers have contributed to the initiation and development of CAM surveys, both historically and for this project. COVID-CAM data were provided by Cancer Research UK who collected the data via Dynata's online survey panels. External Scientific Advisory Group members who advised the statistical analysis plan include: Professor Jamie Brown (University College London), Professor Yoryos Lyratzopoulos (University College London), Dr Katie Robb (University of Glasgow), Dr Christian von Wagner (University College London) and Professor Fiona Walter (University of Cambridge).

**Contributors** KB (responsible for funding acquisition), RC-J, JT, KLW, JW, HDQ-S, KO, GM, MR, JHe, AG contributed to the study design, protocol development, study management and planning. JHe provided lay input to the study design, study implementation and interpretation of the results. MG built the online survey tool and managed the data for the COVID-19 Cancer Help-Seeking and Behaviour Study. VW managed the CRUK COVID-CAM. YM is the CABS study manager. DG and RC-J verified the data and carried out the statistical analysis. YM, HDQ-S, JHu and GMMcC collected and analysed the qualitative data. All authors reviewed the data analyses, contributed to data interpretation and writing of the manuscript, and approved the final version of the submitted manuscript.

**Funding** This study was facilitated by HealthWise Wales, the Health and Care Research Wales initiative, which is led by Cardiff University in collaboration with SAIL, Swansea University. This research comes under the auspices of the Health and Care Research Wales funded Primary and Emergency Care Research Centre (PRIME) (517195) and Wales Cancer Research Centre (517190). DECIPHer and The Centre for Trials Research receive funding from Health and Care Research Wales and Health and Care Research Wales and Cancer Research UK. JW is funded by a Cancer Research UK Career Development Fellowship (C7492/A17219). This work was supported by Economic and Social Research Council as part of UK Research and Innovation's Rapid Response to COVID-19 grant number ES/V00591X/1.

**Disclaimer** The funders had no role in the design, conduct or analyses of this study.

**Competing interests** None declared.

**Patient consent for publication** Not required.

**Ethics approval** Ethical approval was granted by the School of Medicine Research Ethics Committee, Cardiff University (ref 20.68). Informed consent was provided from all participants at recruitment. This study was conducted in accordance with Good Clinical Practice and the Declaration of Helsinki.

**Provenance and peer review** Not commissioned; externally peer reviewed.

**Data availability statement** Data are available on reasonable request. De-identified participant data will be made available to the scientific community with as few restrictions as feasible, whilst retaining exclusive use until the publication of major outputs. Data will be available on a public archive.

includes any translated material, BMJ does not warrant the accuracy and reliability of the translations (including but not limited to local regulations, clinical guidelines, terminology, drug names and drug dosages), and is not responsible for any error and/or omissions arising from translation and adaptation or otherwise.

**ORCID iDs**
Yvonne Moriarty http://orcid.org/0000-0002-7608-4699
Grace M McCutchan http://orcid.org/0000-0002-8079-2540
Jo Waller http://orcid.org/0000-0003-4025-9132
Michael Robling http://orcid.org/0000-0002-1004-036X
Kate Brain http://orcid.org/0000-0001-9296-5748

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
