## [Reviewer comments · BMJ Open]

ARTICLE DETAILS

TITLE (PROVISIONAL)	Cancer symptom experience and help-seeking behaviour during the COVID-19 pandemic in the United Kingdom: a cross-sectional population survey
AUTHORS	Quinn-Scoggins, H; Cannings-John, Rebecca; Moriarty, Yvonne; Whitelock, Victoria; Whitaker, Katriina; Grozeva, Detelina; Hughes, Jacqueline; Townson, Julia; Osborne, Kirstie; Goddard, Mark; McCutchan, GM; Waller, Jo; Robling, Michael; Hepburn, Julie; Moore, Graham; Gjini, Ardiana; Brain, Kate

VERSION 1 – REVIEW

REVIEWER	ZHOU, Qian Huazhong University of Science and Technology Tongji Medical College, School of Medicine and Health Management
REVIEW RETURNED	21-May-2021

GENERAL COMMENTS	On page 6, the author stated that “We conducted a large-scale population survey informed by relevant theory”. I think the author should answer several questions: What theory has applied (based on the reference cited), the purpose to use theory, how the theory guide this study? At the end of the introduction, the author pointed that the gap of the difference in help-seeking behaviors before and after the pandemic. It is better to introduce the background and gap before the last paragraph, and state the purpose and how to fill the gap in this study. this part should be consistent with the main point of this study. The author should put all relevant measures in the survey measures section, but not in other sections or supplements. The measurement of help-seeking behaviors should be listed and cited. In sample size, the author only mentioned the interval, which is only performed by descriptive analysis. It is not appropriate. In the correlated of symptom help-seeking behaviors section, why the author listed these key factors. There are more relevant factors of help-seeking behaviors investigated in previous studies. How about analyzing the barriers and enablers further quantitatively, and merging them into “correlates of symptom help-seeking behavior”?
---

	In the result section, the title of symptom help-seeking behavior is confused. The presentation order should be adjusted, how about 1“symptom prevalence”, 2“symptom help-seeking behavior (should be modified)”, 3“correlates” In table 3, “predict” should be changed. The limitation should be stated more clearly.
--	---

REVIEWER	Mwaka, Amos Makerere University, Medicine
REVIEW RETURNED	23-May-2021

GENERAL COMMENTS	Dear Authors, I am quite impressed with the manuscript; congratulations for the great work. I just have a few comments that perhaps will help improve the quality of the manuscript.  1. Strengths and limitations Bullet three: The tense used may need to be in the past rather than as it is currently. 2. Page 13, line 13/14: You found that remote consultation was not a barrier; I understand that patients (participants) did not find doing remote consultation as a significant issue. Is that data from the qualitative component that could throw some more light as to why patients may have not used that avenue? Or if they successfully used it, then it might be a good point to emphasize. 3. Page 14; lines 27 – 31. The use of the word “enablers” is rather different from the usual and you may want to provide the operational definition and its contextual use here. How does “having a symptom that was bothersome” become an enabling factor for help-seeking? It could drive and or trigger help-seeking in spite of other barriers, but it cannot in itself become an enabler for help-seeking unless the operational definition is provided to guide this contextual use of the word. I appreciate the multiple ways words can often be used. Just like common medicines could be used for uncommon conditions, words can also be used that way; but for such to be useful, the context needs to be provided. I apply the same reasoning to other “enablers” including symptoms that did not go away etc. 4. The qualitative component:  (a) It would be great for you to provide the factors considered in the “purposeful selection” of the participants – the subsample, for the qualitative component. This is well readers in contextualizing the message therefrom. (b) Table S7: This looks like codes and the codes definition rather than themes and themes definitions. The column “Themes identified” would become “Codes”. Reflect over this while putting in mind the bigger picture of approaches to qualitative data analysis. (c) Still table S7; some of the definitions need revision; for example, “The help-seeking interval” – “any discussion or references on why the participant acted when they did regarding the help-seeking or why they waited.” While the “When” connotes an aspect of time, it does not clearly refer to “interval” or duration
--

of time from one point in time to another point. The why is rather indicative of the rationale for acting or not acting at the given time action was taken or not/delayed. A complete review of the code definitions could help improve on the manuscript and point out that the analysis was so done appropriately, given that the codes drive the process of the analysis especially is the codes are developed from the data.

(d) The interview topic guide – S2. The guide looks a revised version and not what was used; it is in the reported speech format, and could not have been used that way with the participants. It would be appropriate to have the wording in the manner directly usable to the participants, rather than in reported version, the 3rd person if you wish to call it that way.

Otherwise, this is a great manuscript.

Regards

VERSION 1 – AUTHOR RESPONSE

Reviewer 1

Comment 1 - On page 6, the author stated that “We conducted a large-scale population survey informed by relevant theory”. I think the author should answer several questions: What theory has applied (based on the reference cited), the purpose to use theory, how the theory guide this study?

Response 1 - We have clarified the use of theory in the Introduction on page 5. Of note the selection of survey measures and framing of qualitative interview topics were guided by health psychology theories including the Model of Pathways to Treatment (ref 13, Scott et al, 2013) and Common Sense Model of Self-Regulation (ref 19, Leventhal et al, 2003).

Comment 2 - At the end of the introduction, the author pointed that the gap of the difference in help-seeking behaviors before and after the pandemic. It is better to introduce the background and gap before the last paragraph, and state the purpose and how to fill the gap in this study. this part should be consistent with the main point of this study.

Response 2 - Thank you for this helpful suggestion. We have altered the Introduction page 5 accordingly.

Comment 3 - The author should put all relevant measures in the survey measures section, but not in other sections or supplements. The measurement of help-seeking behaviors should be listed and cited.

Response 3 - All relevant survey measures have been removed from supplementary materials and moved into the main manuscript and cited as appropriate. These details can now be found in the methods survey measures section under their appropriate sub-headings on pages 6-8.

Comment 4 – In sample size, the author only mentioned the interval, which is only performed by descriptive analysis. It is not appropriate.

Response 4 - This was an oversight. The original study was powered to examine the determinants of self-reported symptom help-seeking interval using multivariable linear regression modelling. The primary outcome was then, a priori to any analysis, changed to a dichotomous outcome (this change was

reflected in the OSF published protocol and statistical analysis plan - <https://osf.io/zxyp3/>). We have therefore now added this detail to the manuscript on pages 8 and 9.

Comment 5 - In the correlated of symptom help-seeking behaviors section, why the author listed these key factors. There are more relevant factors of help-seeking behaviors investigated in previous studies. How about analyzing the barriers and enablers further quantitatively, and merging them into “correlates of symptom help-seeking behavior”?

Response 5 - Thank you for these suggestions. We measured other relevant influences on help-seeking behaviour such as perceived symptom seriousness, perceived risk of cancer and beliefs about cancer in the HealthWise Wales CABS sample, but were not able to measure these constructs in CRUK’s COVID-CAM sample due to restrictions on survey length. We were therefore restricted in the current analyses to those variables for which pooled data from both samples were available. We have amended the methods on page 5 to reflect this.

We also welcome the reviewer’s suggestion to further analyse barriers/prompts to help-seeking, and are planning to analyse these items in more detail and as a further outcome in future analyses of this large dataset.

Comment 6 - In the result section, the title of symptom help-seeking behavior is confused.

The presentation order should be adjusted, how about 1“symptom prevalence”, 2“symptom help-seeking behavior (should be modified)”, 3“correlates

Response 6 - We have adjusted the order of the results and changed the language around help-seeking behaviour throughout the manuscript (pages 9-19). We now present:

- Characteristics of participants
- Symptom prevalence
- Symptom help-seeking
- Correlates of symptom help-seeking
- Help seeking barriers and prompts

Comment 7 - In table 3, “predict” should be changed.

Response 7 - Thank you. This has been changed to ‘Table 3: Unadjusted and adjusted logistic regression models for self-reported GP contact in participants who experienced at least one potential cancer symptom, UK, August-September 2020...’ (page 16).

Comment 8 - The limitation should be stated more clearly.

Response 8 - We have altered the relevant sentence in the Discussion on page 24 to state the study limitations more clearly.

Reviewer 2

Comment 1 - Strengths and limitations. Bullet three: The tense used may need to be in the past rather than as it is currently.

Response - Thank you for this comment. We have amended the tense in this bullet point to the past tense now reading “Data collection occurred between August and September 2020 and thus reflects the first lockdown period in the UK” (page 3).

Comment 2 - Page 13, line 13/14: You found that remote consultation was not a barrier; I understand that patients (participants) did not find doing remote consultation as a significant issue. Is that data from the qualitative component that could throw some more light as to why patients may have not used that avenue? Or if they successfully used it, then it might be a good point to emphasize.

Response 2 - Thank you for the comment and consideration of this important issue. As exploration of use and/or experience of remote consultations was not the primary aim of the qualitative work, we have little further data to expand here. Due to the numbers in the qualitative sample, we are reluctant to over-emphasise findings in this regard. We have, however, suggested that further research in this area is needed with participants who have a range of digital literacy and health motivation (page 25). Both of which are likely high in our participant cohort due to the sampling methods used.

Comment 3 - Page 14; lines 27 – 31. The use of the word “enablers” is rather different from the usual and you may want to provide the operational definition and its contextual use here. How does “having a symptom that was bothersome” become an enabling factor for help-seeking? It could drive and or trigger help-seeking in spite of other barriers, but it cannot in itself become an enabler for help-seeking unless the operational definition is provided to guide this contextual use of the word. I appreciate the multiple ways words can often be used. Just like common medicines could be used for uncommon conditions, words can also be used that way; but for such to be useful, the context needs to be provided. I apply the same reasoning to other “enablers” including symptoms that did not go away etc.

Response 3 - Thank you for the consideration given to this. We agree that the terminology of ‘enablers’ may not be relevant for some of the factors. In recognition of this we have changed ‘enablers’ to ‘prompts’ throughout the manuscript where appropriate, which we believe better represents the operational definition.

Comment 4 - The qualitative component:

(a) It would be great for you to provide the factors considered in the “purposeful selection” of the participants – the subsample, for the qualitative component. This is well readers in contextualizing the message therefrom.

(b) Table S7: This looks like codes and the codes definition rather than themes and themes definitions. The column “Themes identified” would become “Codes”. Reflect over this while putting in mind the bigger picture of approaches to qualitative data analysis.

(c) Still table S7; some of the definitions need revision; for example, “The help-seeking interval” – “any discussion or references on why the participant acted when they did regarding the help-seeking or why they waited.” While the “When” connotes an aspect of time, it does not clearly refer to “interval” or duration of time from one point in time to another point. The why is rather indicative of the rationale for acting or not acting at the given time action was taken or not/delayed. A complete review of the code definitions could help improve on the manuscript and point out that the analysis was so done appropriately, given that the codes drive the process of the analysis especially is the codes are developed from the data.

(d) The interview topic guide – S2. The guide looks a revised version and not what was used; it is in the reported speech format, and could not have been used that way with the participants. It would be appropriate to have the wording in the manner directly usable to the participants, rather than in reported version, the 3rd person if you wish to call it that way.

Response 4 - We thank the reviewer for the comment and direct them to page 8 in the method section where this detail is provided. Factors for qualitative purposeful sampling included age, gender and symptom experience.

Thank you for the two comments on the supplementary table. Please find the below changes made to the table and elsewhere throughout the manuscript to increase clarity across the methods and definitions used in the qualitative work in response to points 4 (a) and (b):

We have changed the title of supplementary table S5 to 'Definitions of codes identified during the qualitative interviews with participants and mapping of these to the themes of symptom experiences, fear of help-seeking and experiences of help-seeking'

An additional column has been added to the left-hand side of the table to show how codes identified were extrapolated and mapped into the themes presented in main text

Column headings for columns 2 and 3 have been changed recognising that these relate to codes and not themes – they now read 'Codes' and 'Code definitions'

Code definitions have been revised to provide further clarity where appropriate.

The code of 'The help-seeking interval' has now been merged with the code 'Actions taken, or not taken, due to changes in health or body' on reflection of your comment on the meaning of interval and time parameters associated

To further clarify the analysis method additional detail has been added to the methods section on qualitative interviews page 8 of the main manuscript highlighting that the coding was conducted both inductively and deductively using a priori codes. A priori codes were derived from previous research in this area and the topic guide used to conduct the semi-structured interviews for this study.

Thank you for this suggestion. We have now included the expanded semi-structured topic guide in the supplementary materials that was used with participants during the interviews (Supplementary material S3)